# Stand-Level Biomass and Leaf Trait Models for Young Naturally Regenerated Forests of European Hornbeam

Bohdan Konôpka [1,2], Vlastimil Murgaš [1], Vladimír Šebeň [1,*], Jozef Pajtík [1] and Katarína Merganičová [3]

1   National Forest Centre, Forest Research Institute, SK-960 01 Zvolen, Slovakia; bohdan.konopka@nlcsk.org (B.K.); vlastimil.murgas@nlcsk.org (V.M.); jozef.pajtik@nlcsk.org (J.P.)
2   Faculty of Forestry and Wood Sciences, Czech University of Life Sciences Prague, Kamýcká 129, CZ-165 000 Prague, Czech Republic
3   Slovak Academy of Sciences, Institute of Landscape Ecology, SK-949 01 Nitra, Slovakia; katarina.merganicova@forim.sk
*   Correspondence: vladimir.seben@nlcsk.org; Tel.: +421-455-314-181

**Abstract:** European hornbeam (*Carpinus betulus* L.) is a tree species widely distributed in Europe and the Asian part of the Near East. However, since European hornbeam is not very attractive for commercial purposes, scientific interest in this species has been rather sparse. Our study focused on dense young (up to 10 years old) European hornbeam stands originating from natural regeneration from seeds in Slovakia because in future the importance of this species may increase due to the climate change. We combined previously constructed tree-level biomass models, data on basic leaf traits, i.e., weight and area, and measurements from thirty plots located at ten different sites across Slovakia to construct stand-level allometric relations of the biomass stock in tree components, i.e., leaves, branches, bark, stem under bark and roots, to mean stand diameter at stem base, i.e., at the ground level. Moreover, we calculated and modelled leaf characteristics, namely the specific leaf area (SLA), leaf area ratio (LAR) and leaf area index (LAI), at a stand level. The total tree biomass stock including all tree components ranged between 0.75 and 13.63 kg per m$^2$, out of which the biomass of stem with bark was from 0.31 to 8.46 kg per m$^2$. The biomass models showed that the contribution of roots (omitting those with a diameter under 2 mm) decreased with the increasing mean stand diameter at stem base, whereas the opposite pattern was observed for branches and stem biomass. Further, we found that the mean stand diameter at stem base was a good predictor of both LAR and LAI. The results indicated the high photosynthetic efficiency of European hornbeam leaves per one-sided surface leaf area. Moreover, the growth efficiency (GE), expressed as the biomass increment of woody parts per leaf area unit, of young European hornbeam trees was high. The models proved a close positive linear correlation between LAI and stand biomass stock that may be used for estimating the biomass in young stands from LAI that can be measured using non-destructive terrestrial or aerial methods. The results further indicated that young stands may sequester a non-negligible quantity of carbon; therefore, they should not be omitted from local or country-wide estimates of carbon stocks in forest vegetation.

**Keywords:** *Carpinus betulus* L.; dense stands; tree components; foliage traits; allometric relations

## 1. Introduction

European (sometimes "common") hornbeam (*Carpinus betulus* L.) is a medium-sized deciduous hardwood tree species reaching a maximum height of about 25 m [1]. The species has a wide distribution range from southern Europe (except from the Iberian Peninsula), through Central Europe up to the very southern regions of England and Sweden; eastwards, the European hornbeam occurs around the Black Sea, reaching the Caucasus and northern parts of Iran [1]. In Slovakia, the European hornbeam is very common and abundant, especially in the central and southern regions of the country [2]. The species

occurs on lowlands, wold and low hills, from the lowest altitudes to about 800 m above sea level. European hornbeam often creates mixed stands with oaks (*Quercus* spp.) and European beech (*Fagus sylvatica* L.); it occasionally grows on rocky sites together with some other broadleaved species such as *Acer pseudoplatanus* L., *Fraxinus excelsior* L. and *Tilia platyphyllos* Scop. [3]. This species can create coppice stands, i.e., it frequently reproduces from sprouts [2]. However, the Slovak forestry practice has continually decreased the proportion of coppice hornbeam stands since the middle of the last century. Thus, recent hornbeam forests originate mostly from natural seed reproduction that should have better wood quality than coppices [4]; however, see also Vollmuth [5]. The latest Slovak national inventory in 2015 and 2016 showed that the species covered an area of about 190 thousand ha, was the fourth most common species based on the spatial proportion, was ranked sixth when considering the stand stock and occurred in 32% of the inventory plots [4].

European hornbeam is not used in the wood-processing industry because its wood is heavy, prone to warping and moisture change and has an inexpressive grey colour and unattractive texture. Although its mechanical strength is high and thermal expansion is low, it is hygroscopic and rather vulnerable to biological agents [6]. Since European hornbeam is not very attractive for commercial purposes, scientific interest in this species has been less intensive than for other common tree species. On the other hand, since this species is rather frequent in Europe, it could be important due to its other ecological roles such as in erosion control, improvement of chemical and mechanical soil properties, etc. (see for instance [7,8]), or for carbon sequestration.

Thus, suitable tools for the estimation of its biomass are needed. Several allometric models were derived for European hornbeam within Europe, for instance in Albania [9], Belgium [10] and Germany [11] and within the Near East in Iran [12,13] and Turkey [14]. However, all mentioned works focused exclusively on coppice stands. For the conditions of the Western Europe, Annighöfer et al. [15] constructed aboveground biomass equations for seedlings and saplings of 19 species including European hornbeam. Summarizing the available literature, we conclude that allometric relations for European hornbeam in the Western Carpathians and models for its belowground tree parts are still missing. For the conditions of Slovakia, allometric models for all tree components (leaves, branches, stem and roots) of eleven tree species in young growth stages were constructed and published in a monograph by Pajtík et al. [16]. The work also presents biomass models for European hornbeam but only at a tree level.

Although foliage is one of the tree parts with little contribution to total tree biomass, it is an extremely physiologically active organ, especially if we focus on its surface area [17]. Several works [18–20] implement the ratio between the foliage area and the total plant dry biomass (leaf area ratio; LAR) to link it to ecological and production processes. For similar purposes, specific leaf area (ratio between leaf area and dry leaf mass; SLA) has been studied, especially as an indicator of the adaptation strategy in a variety of plants [21]. Canopy leaf area serves as a dominant physical driver of forest production, transpiration, energy exchange and other physiological attributes related to a variety of ecosystem processes; hence, it is an important integrant of ecological studies [22]. Therefore, the leaf area index (amount of leaf area in the canopy per unit ground area; LAI) is the most useful canopy parameter [23]. Our literature search suggested that results on the LAR, SLA and LAI of European hornbeam were still sparse. On the currently available information on European hornbeam, the LAR and SLA are represented at either the leaf or tree level [24] but nothing is available for the stand level.

The aim of this study was to construct stand-level allometric relations for biomass stock in tree components of young European hornbeam stands based on the mean diameter at stem base. Moreover, we focused on deriving models that express leaf characteristics at a stand level, namely SLA, LAR and LAI. Finally, we modelled the stand biomass stock using the LAI as a predictor.

## 2. Materials and Methods

### 2.1. Stand Selection and Tree Sampling

The selection of stands was performed using the current national database of forest stands derived from the information in forest management plans (see also http://gis.nlcsk.org/lgis/, accessed on 20 April 2023). The main criteria for the selection of sampled stands were as follows: the share of European hornbeam in tree species composition had to be 100%; stands had to originate exclusively from natural seed regeneration; their canopy had to be fully closed (equal to or approaching 100%); and their mean stand age had to be up to 10 years. Moreover, the stands had to grow at moderately fertile sites. Subsequently, we performed field surveys and selected ten stands for further analyses. The selected stands were distributed across the Slovak territory in a west to east direction from the Malé Karpaty Mts. to the Nízke Beskydy Mts (Figure 1). The altitudes of the sampled sites were between 295 m and 567 m above sea level (Table 1).

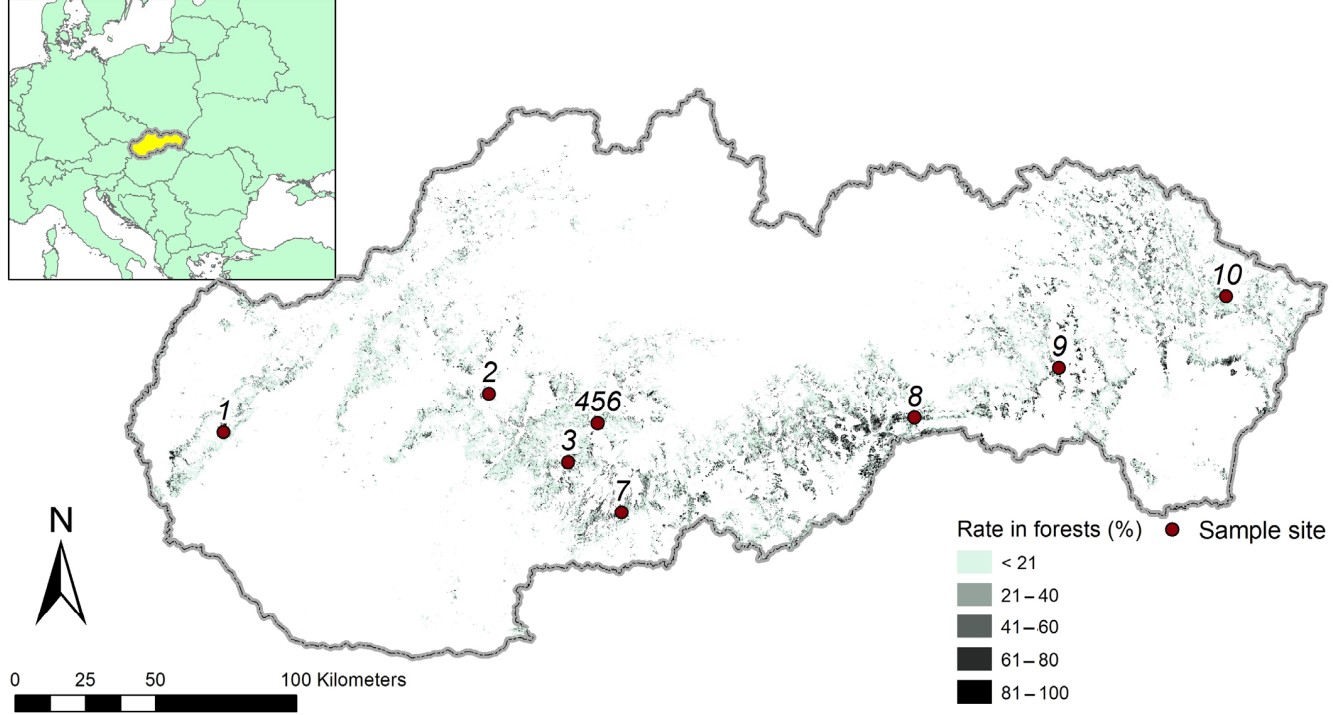

**Figure 1.** Localization of sampled sites marked with specific codes. Background colours illustrate the occurrence of European hornbeam in Slovak forests (according to Šebeň [4]) with the specification of its contribution to tree species composition in the stands. See Table 1 for the names of sampled sites related to the specific codes.

**Table 1.** Basic characteristics of the sampled sites of European hornbeam in Slovakia.

| Site Code | Site Name | Altitude | N Latitude | E Longitude | Aspect | Soil | Bedrock |
|---|---|---|---|---|---|---|---|
| | | (m a.s.l.) | (°) | (°) | | | |
| 1 | Píla | 313 | 48.3941 | 17.2944 | S | Typical paternia | Alluvium |
| 2 | Rudica | 475 | 48.5935 | 18.5497 | W | Mesotropic cambisols | Andesites |
| 3 | Antol | 516 | 48.3955 | 18.9564 | W | Ilimerised soil | Clay loess |
| 4 | Dol. Breziny | 387 | 48.5274 | 19.0820 | N | Mesotropic cambisoils | Andesitic tuff |
| 5 | Šariny | 432 | 48.5271 | 19.0816 | NE | Mesotropic cambisoils | Andesites |
| 6 | Hor. Breziny | 451 | 48.5361 | 19.0796 | E | Mesotropic cambisoils | Andesites |
| 7 | Cerovo | 560 | 48.2475 | 19.2295 | SW | Mesotropic cambisoils | Andesitic tuff |
| 8 | Soroška | 567 | 48.6109 | 20.6057 | NW | Moder rendzinas | Limestones |
| 9 | Budimír | 295 | 48.7938 | 21.2916 | SW | Ilimerised soil | Clay loess |
| 10 | Zubné | 350 | 49.0459 | 22.0874 | N | Ilimerised soil | Sandstones |

In total, 200 European hornbeam trees from ten selected forest stands were destructively sampled and measured (Table A1 in Appendix A). At each site, 20 trees were chosen to cover the entire tree height range present in each stand. Sampling was performed in the second part of the growing season in 2017 when shoots and leaves were fully developed. The sampled trees were selected randomly from healthy trees, thus avoiding damaged, deformed or atypically shaped trees and trees growing on forest stand edges. Severely suppressed individuals showing initial symptoms of crown dieback were also avoided (see also [16] for more information). The sampled trees were subjected to further measurements and laboratory treatments to determine the dry mass of the individual tree components, i.e., leaves, branches, bark on stem, stem under bark and roots (omitting those under 2 mm in diameter). The detailed description of the laboratory work as well as of the mathematical modelling is given in previous papers (e.g., [16,25,26]). Here, we implemented allometric relations already published by Pajtík et al. [16] (see also Table A2 in Appendix A).

During the tree sampling, nine leaves along the vertical profile of each tree crown were randomly selected and cut with a pair of scissors. Hence, about 1800 leaves were harvested from the entire set of sampled trees. Leaves were packed in marked paper envelopes and individually scanned (scanner Epson Expression 10000 XL) in the laboratory while still fresh. Then, each leaf was dried in an oven to a constant weight (under 95 °C for 24 h). Dried leaves were weighed using a precise laboratory scale (precision ±0.0001 g). The area of each leaf was calculated from the scanned images using the Easy Leaf Area programme [27].

In parallel with tree sampling, research plots were subjected to dendrometric tree measurements. Specifically, three circular plots each with a radius between 0.5 m and 1.5 m were established in each stand, totalling 30 plots (Table A3 in Appendix A). The length of the radius was chosen to include at least 40 trees within each plot. The plots were located in the same stands from which the trees were destructively sampled but we avoided the places with harvested trees. Then, tree heights and diameters at stem base (diameter $D_0$ hereinafter) of all trees inside the circular plots were measured. Tree heights were measured with a telescopic measure (precision of ±1 cm); diameters $D_0$ were measured with a digital calliper (±0.1 mm).

### 2.2. Calculations at Leaf, Tree and Stand Levels

Individual tree measurements were processed at the level of individual circular plots. The arithmetic means and the standard deviations of tree heights and $D_0$ diameters were calculated at each plot. The number of trees per $m^2$ was calculated as the sum of all trees in the plot divided by the plot area. The mean stem-base basal area (more precisely: the sum of tree sections per $m^2$ of stand area calculated from stem diameters measured at stem bases, i.e., at the ground level) at each plot was calculated in the same way using $D_0$.

The biomass stocks of the individual tree components from each measured tree were calculated using the previously developed local allometric models [16] (see also Table A2 in Appendix A). The input variables were tree height (in metres) and diameter $D_0$ (in millimetres). The biomass stocks of leaves, branches, barks and stems under bark in grams were calculated separately, and their sum represented the aboveground tree biomass. Similarly, the belowground tree biomass was calculated using the mentioned allometric models. By summing the aboveground and belowground biomass, we obtained the total tree biomass. The total dry biomass at each plot was calculated as the sum of the biomass of all trees. The biomass per unit area (g per $m^2$) was calculated by dividing the total biomass per plot with the plot area.

Nonlinear height–diameter functions with a single predictor variable have been commonly used to describe tree height and diameter relationships. We used a simplified version of the Näslund function (e.g., [28]) to describe this relationship:

$$H = \frac{D_0^2}{b_0 + b_1 D_0 + b_2 D_0^2} \tag{1}$$

where $b_0$, $b_1$ and $b_2$ are the parameters to be estimated, $H$ is tree height (m) and $D_0$ is diameter at stem base (mm).

The general form of biomass models for individual tree components (leaves, branches, bark, stem wood, roots, aboveground and total biomass) at a plot level was as follows:

$$B_i = b_0 D_0^{b_1} \tag{2}$$

where $B_i$ is the dry biomass, i.e., weight of matter with zero water content (g per m$^2$), $D_0$ is mean stand diameter at stem base (mm) and $b_1$ and $b_1$ are parameters to be estimated. The biomass distribution of tree components depending on the mean diameter $D_0$ is shown using fitted values from the biomass models derived at the plot level.

Specific leaf area (SLA; in cm$^2$ per g) was calculated as the ratio between the measured leaf area (LA; in cm$^2$) and the leaf weight ($w_f$; in grams), first at a leaf level, then an average value was derived at a tree level (sampled trees). Afterwards, the average SLA values were calculated from the sampled trees for each site. Afterwards, LA at a tree level (in mm$^2$) was calculated by multiplying the mean site-specific value of the SLA with the dry leaf biomass of an individual tree. The leaf area ratio (LAR; cm$^2$ per g) was calculated as the ratio between the tree leaf area (LA) and the total tree biomass.

Subsequently, we derived regression models using a power function or a linear function for predicting the leaf area (LA), leaf area ratio (LAR), leaf area index (LAI; m$^2$ per m$^2$) and total dry biomass ($B_{Total}$; in g) at an individual tree or a plot level as follows:

$$\text{LA} = b_0 w_f^{b_1} \tag{3}$$

and

$$\overline{\text{LA}} = b_0 + b_1 \overline{w}_f \tag{4}$$

where $\overline{\text{LA}}$ is the mean leaf area (in m$^2$) and $\overline{w}_f$ is the mean leaf weight (in g) at the plot level.

$$\text{LAR} = b_0 D_0^{b_1} \tag{5}$$

$$\text{LAI} = b_0 D_0^{b_1} \tag{6}$$

$$B_{Total} = b_0 + b_1 \text{LAI} \tag{7}$$

We used NSE (Nash–Sutcliffe efficiency; [29]), RMSE (root mean squared error), PBIAS (percent bias), IOA (index of agreement), AIC (Akaike information criterion) and the Bayesian information criterion (BIC) to evaluate the quality of the models.

$$\text{NSE} = 1 - \frac{\sum_{i=1}^{n}(y_i - \hat{y}_i)^2}{\sum_{i=1}^{n}(y_i - \overline{y}_i)^2} \tag{8}$$

$$\text{RMSE} = \sqrt{\text{MSE}} = \sqrt{\frac{1}{n}\sum_{i=1}^{n}(y_i - \hat{y}_i)^2} \tag{9}$$

where:

SSE—sum of squares error;
SST—sum of squares total;
$\hat{y}$—predicted value of $y$;
$\overline{y}$—mean value of observed $y$.

$$\text{PBIAS} = 100 \times \frac{\sum_{i=1}^{n}(y_i - \hat{y}_i)}{\sum_{i=1}^{n}(y_i)} \tag{10}$$

$$IOA = 1 - \frac{\sum_{i=1}^{n}(y_i - \hat{y}_i)^2}{\sum_{i=1}^{n}(|\hat{y}_i - \overline{y}| + |y_i - \overline{y}|)^2} \tag{11}$$

NSE is a normalized statistic that determines the relative magnitude of the residual variance compared with the measured data variance [29]. NSE indicates how well the plot of observed versus simulated data fits the 1:1 line. NSE = 1 corresponds to a perfect match of the model to the observed data. NSE = 0 indicates that model predictions are as accurate as the mean of the observed data. Inf < NSE < 0 indicates that the observed mean is a better predictor than the model.

RMSE measures the average difference between values predicted by a model and the actual values. It can be interpreted as the standard deviation of the error. Lower values for RMSE indicate better model performance.

PBIAS measures the average tendency of the simulated data to be larger or smaller than their observed counterparts [30]. The optimal value of PBIAS is 0, and low-magnitude values indicate accurate model predictions. In this study, positive values indicate model underestimation bias and negative values indicate model overestimation bias.

IOA was developed by Willmott [31] as a standardized measure of the degree of numerical model prediction error and varies between 0 and 1. A model performs best if the value of IOA is 1.

All statistical analyses were performed using R programming language 4.2.2 (R Core [32]) and graphically visualized using the ggplot2 3.4.0 [33] package of R. The 95% confidence and prediction intervals for the fitted allometric models were computed with the predFit function in the investr package version 1.4.2 (see https://scholar.afit.edu/facpub/174/, accessed on 20 April 2023).

### 3. Results

Our results showed very close relations between diameter $D_0$ and tree height at the tree level as well as between mean diameter $D_0$ and mean tree height at the plot level (see intervals of confidence and prediction along fitting curves in Figure 2 and values of statistical criteria in Tables 2 and A4 of Appendix A). Naturally, a closer relationship was found at the plot level, since it is based on means of characteristics that are less sensitive to extreme values (outliers). Although in this work other plot-level models (i.e., biomass stocks of individual tree components, LAR and LAI) were based on mean diameter $D_0$, the relationship between $D_0$ and mean tree height provides users with an opportunity to calculate biomass and leaf characteristics also using mean tree height as a predictor.

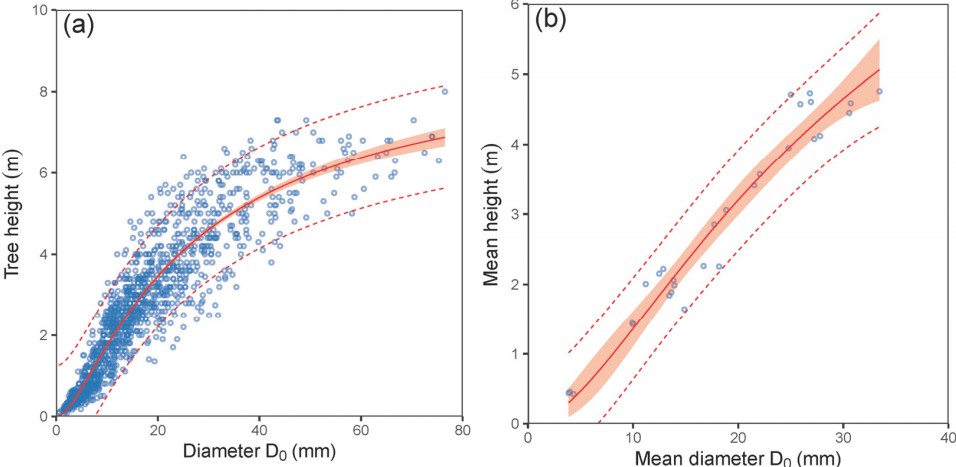

**Figure 2.** Relationship between tree diameter at stem base $D_0$ and tree height (**a**) and between mean diameter at stem base $D_0$ and mean height (plot level; (**b**)) derived from measured European hornbeam trees (characteristics of allometric models are shown in Table 2; statistic criteria are in Table A4). The red belts indicate 95% confidence intervals, and dashed red lines denote 95% prediction intervals.

**Table 2.** Mathematical models describing relationships between tree diameter at stem base $D_0$ (in mm) and tree height ($H$; in m) and between mean diameter at stem base $D_0$ and mean height ($H$) at the plot level for European hornbeam. The abbreviations represent: $b_0$, $b_1$, $b_2$—regression coefficients, S.E.—standard error, $p$—$p$ value. See Table A4 in the attachment for other statistic criteria.

| Related Variables | Equation | $b_0$ | S.E. | $p$ | $b_1$ | S.E. | $p$ | $b_2$ | S.E. | $p$ |
|---|---|---|---|---|---|---|---|---|---|---|
| $H$ vs. $D_0$—tree level | (1) | 20.763 | 2.275 | <0.001 | 2.582 | 0.222 | <0.001 | 0.108 | 0.005 | <0.001 |
| Mean $H$ vs. mean $D_0$—plot level | (1) | 38.664 | 25.162 | 0.136 | 2.638 | 2.597 | 0.319 | 0.084 | 0.062 | 0.187 |

The results suggested that the mean stem base diameter was a suitable predictor for estimating the total tree biomass stock as well as the biomass of individual tree components at the plot level (Figure 3; Table 3). The fitted values for instance showed that a hornbeam plot with a mean diameter $D_0$ of 30 mm had biomass stocks of about 962, 1617, 872, 5981 and 1616 g per m$^2$ in foliage, branches, bark, stem under bark and roots, respectively. Hence, the total tree biomass stock in the stands with the mentioned mean stem base diameter was about 11,050 g per m$^2$. The models on biomass stock in individual tree components showed the differences between the components not only in their absolute values but also in their rates with the changing mean diameter $D_0$. In addition, the results indicated changes in the proportions of tree components to total tree biomass stock with stand size (Figure 4). Specifically, whereas the proportion of roots decreased, the proportion of branches and mainly of stem biomass increased with the mean diameter $D_0$ of plot. The largest portion of tree biomass was allocated to stem. On the other hand, the smallest portion was found in leaves, whereas their contribution was rather stable across all plots.

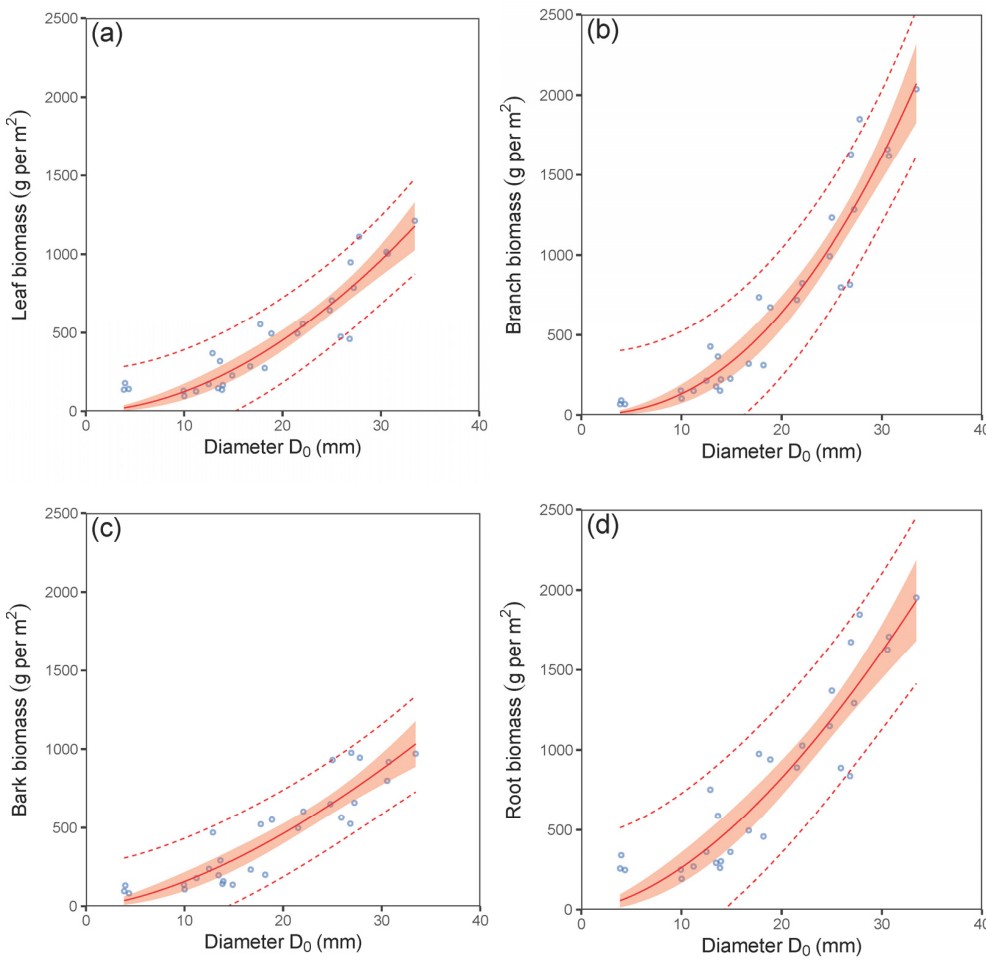

**Figure 3.** *Cont.*

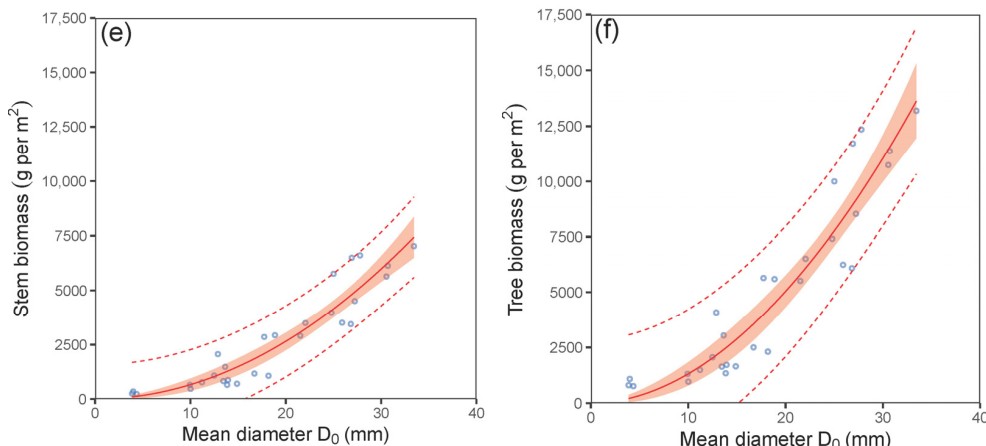

**Figure 3.** Relations between mean diameter at stem base $D_0$ and leaf biomass (diagram (**a**)), branch biomass (**b**), bark biomass (**c**), root biomass (**d**), biomass of stem under bark (**e**) and total tree biomass (**f**) in European hornbeam plots (characteristics of allometric models are shown in Table 3; statistic criteria are in Table A4). The red belts indicate 95% confidence intervals, and dashed red lines denote 95% prediction intervals.

**Table 3.** Allometric models describing relationships between mean diameter at stem base $D_0$ (in mm) and biomass (in g) of the tree components of European hornbeam at the plot level. The abbreviations are explained in the caption of Table 2. See Table A4 in the attachment for other statistic criteria.

| Related Variables | (Equation) | $b_0$ | S.E. | $p$ | $b_1$ | S.E. | $p$ |
|---|---|---|---|---|---|---|---|
| leaf biomass vs. mean $D_0$ | (2) | 1.716 | 1.099 | 0.130 | 1.861 | 0.195 | <0.001 |
| branch biomass vs. mean $D_0$ | (2) | 0.688 | 0.478 | 0.161 | 2.282 | 0.209 | <0.001 |
| bark biomass vs. mean $D_0$ | (2) | 4.318 | 2.630 | 0.112 | 1.560 | 0.187 | <0.001 |
| root biomass vs. mean $D_0$ | (2) | 5.768 | 3.409 | 0.102 | 1.657 | 0.181 | <0.001 |
| stem biomass vs. mean $D_0$ | (2) | 6.604 | 4.424 | 0.147 | 2.002 | 0.203 | <0.001 |
| total biomass vs. mean $D_0$ | (2) | 15.254 | 9.636 | 0.125 | 1.936 | 0.192 | <0.001 |

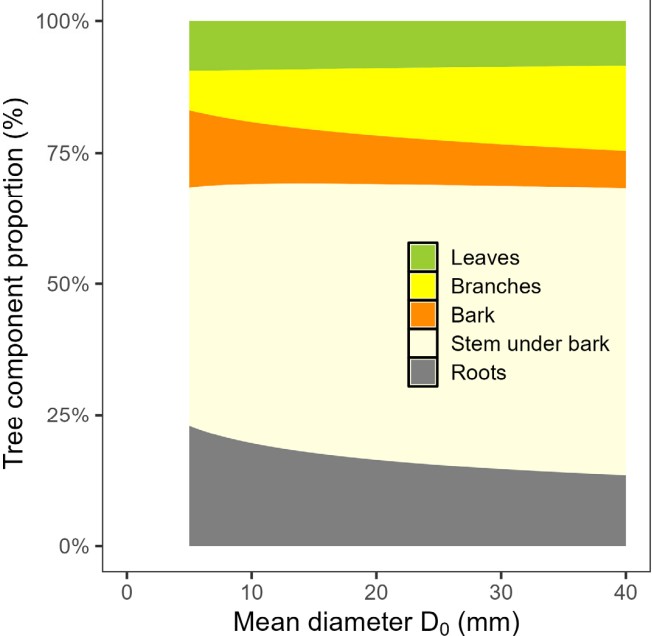

**Figure 4.** Contributions of tree components, specifically leaf, branches, bark, stem under bark and roots, to total tree biomass along the range of mean diameter at stem base $D_0$ for European hornbeam plots.

Our analyses further showed a close relationship between leaf weight and leaf area (Figure 5). Although this relationship was nonlinear (expressed by a power function) at the leaf level, it was clearly linear at the stand level. The SLA at the stand level varied from 46.4 to 80.4 cm$^2$ per g between the sites (Table 4). Although the differences between the stands seemed high, they were not statistically different (ANOVA; $p > 0.05$). The values for the LAR derived from the mean diameter $D_0$ of plots decreased with increasing values of $D_0$ following a nonlinear pattern (Figure 6a; Table 5). The fitted curve indicated that the decreasing rate of the LAR with diameter $D_0$ was steeper in smaller stands ($D_0$ under 10 mm) than in bigger ones. A similar trend in LAR was also found at the plot level (Figure 6b). On the other hand, the opposite pattern to that of the LAR was found for the LAI (Figure 7; Table 5). Here, the LAI increased with mean plot diameter $D_0$ nonlinearly. Finally, we modelled a relationship between the LAI and tree biomass stock for 30 plots, which showed a close steep linear relationship (Figure 8; Table 5).

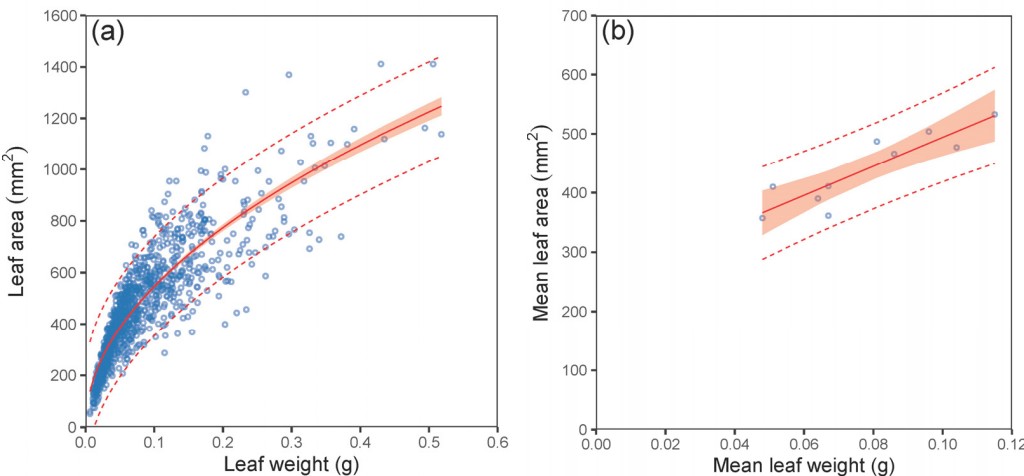

**Figure 5.** Relationship between leaf weight and leaf area of European hornbeam at the tree level (**a**) and mean leaf weight and mean leaf area at the stand level (**b**) (characteristics of allometric models are shown in Table 5; statistic criteria are in Table A4). The red belts indicate 95% confidence intervals, and dashed red lines denote 95% prediction intervals.

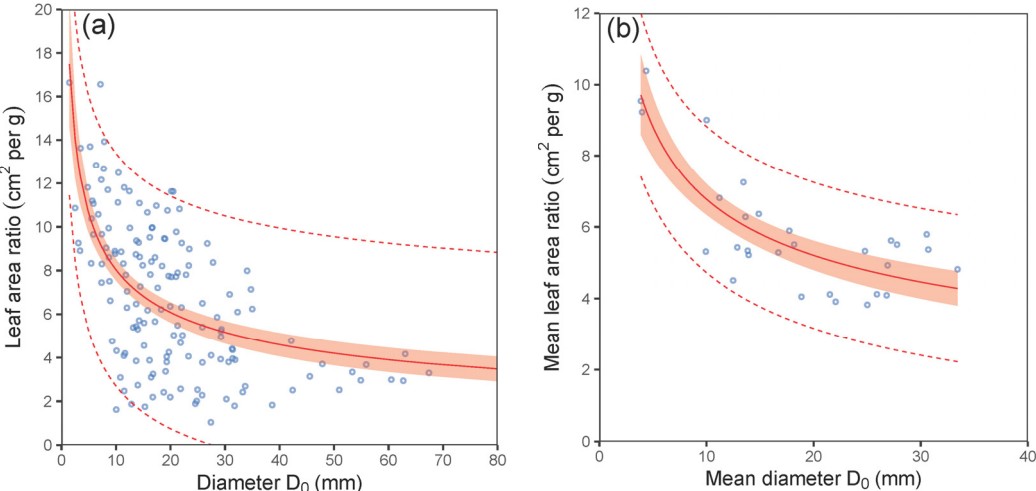

**Figure 6.** Relationship between tree diameter at stem base $D_0$ or mean diameter at stem base $D_0$ and mean leaf area ratio of European hornbeam (LAR) at both the tree (diagram (**a**)) and plot (**b**) levels (characteristics of allometric models are shown in Table 5; statistic criteria are in Table A4). The red belts indicate 95% confidence intervals, and dashed red lines denote 95% prediction intervals.

**Table 4.** Basic leaf characteristics, i.e., leaf area, leaf weight and specific leaf area (SLA) of European hornbeam trees at sampled sites (mean values and standard deviations are shown).

| Site Code | Site Name | Leaf Area | Leaf Weight | SLA |
|:---:|:---:|:---:|:---:|:---:|
| | | (cm$^2$) | (g) | (cm$^2$ per g) |
| 1 | Píla | 5.04 (2.34) | 0.096 (0.072) | 52.5 (23.4) |
| 2 | Rudica | 4.10 (1.82) | 0.051 (0.032) | 80.4 (32.2) |
| 3 | Antol | 3.90 (2.27) | 0.064 (0.037) | 60.9 (27.0) |
| 4 | Dol. Breziny | 3.57 (1.17) | 0.048 (0.029) | 56.3 (21.8) |
| 5 | Šariny | 4.66 (1.99) | 0.086 (0.041) | 54.2 (24.3) |
| 6 | Hor. Breziny | 3.61 (1.63) | 0.067 (0.053) | 53.9 (22.8) |
| 7 | Cerovo | 4.87 (1.25) | 0.081 (0.053) | 60.1 (26.5) |
| 8 | Soroška | 5.33 (2.04) | 0.115 (0.072) | 46.4 (19.7) |
| 9 | Budimír | 4.11 (1.89) | 0.067 (0.051) | 61.3 (31.5) |
| 10 | Zubné | 4.77 (2.40) | 0.104 (0.081) | 45.9 (23.6) |

**Table 5.** Mathematical models describing the relationships between leaf characteristics (leaf area—LA, expressed in cm$^2$; leaf weight—LW, in g; leaf area ratio—LAR, in cm$^2$ per g; and leaf area index—LAI, in m$^2$ per m$^2$) and total tree biomass ($B_{Total}$ in g) of European hornbeam considering specific levels: leaf, tree, plot or stand. The abbreviations are explained in the caption of Table 2. See Table A4 in the attachment for other statistic criteria.

| Related Variables | (Equation) | $b_0$ | S.E. | $p$ | $b_1$ | S.E. | $p$ |
|:---|:---:|:---:|:---:|:---:|:---:|:---:|:---:|
| LA vs. LW—leaf level | (3) | 1734.740 | 33.090 | <0.001 | 0.502 | 0.008 | <0.001 |
| LA vs. LW—stand level * | (4) | 247.971 | 35.780 | <0.001 | 2459.935 | 443.150 | <0.001 |
| LAR vs. $D_0$—tree level | (5) | 20.315 | 1.963 | <0.001 | −0.403 | 0.038 | <0.001 |
| LAR vs. $D_0$—plot level | (5) | 16.241 | 1.787 | <0.001 | −0.380 | 0.043 | <0.001 |
| LAI vs. $D_0$—plot level | (6) | 0.010 | 0.008 | 0.259 | 1.851 | 0.263 | <0.001 |
| $B_{Total}$ vs. LAI—plot level * | (7) | −91.126 | 293.560 | 0.759 | 1961.813 | 90.080 | <0.001 |

Explanatory note: * The models are applicable only for values within the ranges of our measured data.

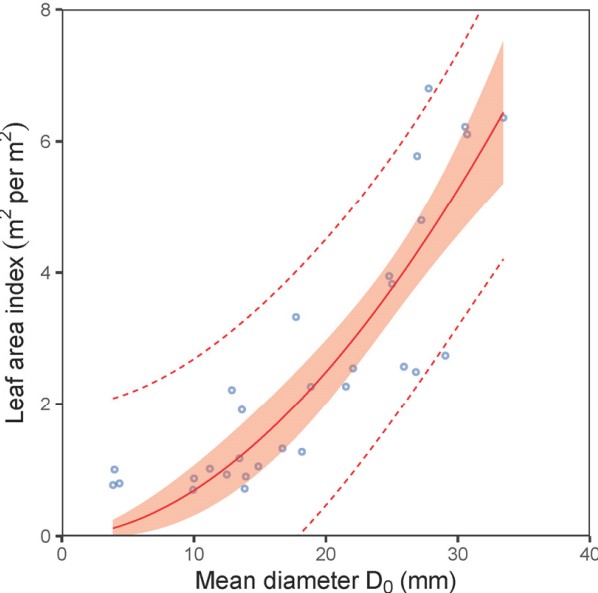

**Figure 7.** Relationship between mean diameter at stem base $D_0$ and leaf area index (LAI) of European hornbeam plots (characteristics of allometric models are shown in Table 5; statistic criteria are in Table A4). The red belt indicates 95% confidence interval, and dashed red lines denote 95% prediction interval.

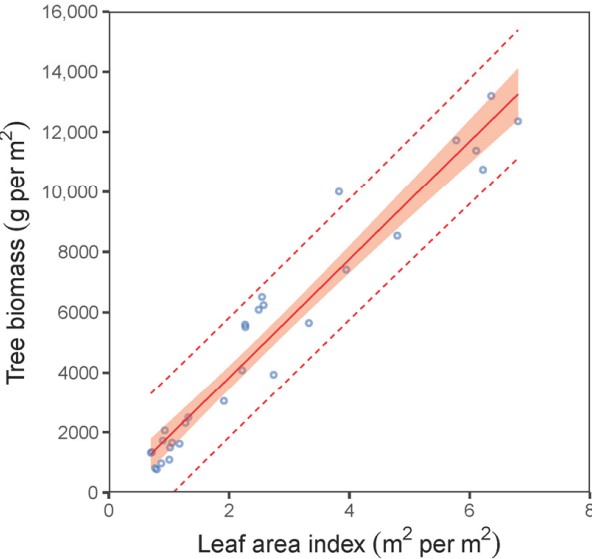

**Figure 8.** Relationship between leaf area index (LAI) and total tree biomass for European hornbeam plots (characteristics of allometric models are shown in Table 5; statistic criteria are in Table A4). The red belt indicates 95% confidence interval, and dashed red lines denote 95% prediction interval.

## 4. Discussion

### 4.1. Stand Biomass—Quantity, Structure and Carbon Context

Our results at the plot level showed that young hornbeam stands (aged up to 10 years) contained total tree biomass up to 125 tons per ha, i.e., over 60 tons of carbon per ha. Unfortunately, we could not find any stand-level biomass models for European hornbeam in the literature. Our previous study in young European beech stands (unpublished data) in central Slovakia showed that biomass stocks in 5- and 10-year-old stands were 58 and 103 tons per hectare, respectively. In our hornbeam stands, the stem component contributed to the total biomass most, whereas its percentage increased with the mean stem base diameter and reached 50% in stands with a mean diameter of 35 mm. According to the Slovak national inventory [4], the mean biomass stock in Slovak forests is about 250 t per ha. Although the inventory provides rather rough biomass estimates, the values indicate that our plots, characterized with the greatest value for mean diameter (33.4 mm), represented approximately half of the average stock in Slovakia. This seems a rather surprising finding since our results originated from young stands (up to 10-year-old). On the other hand, we should note that our hornbeam stands were very dense—with full canopy closure— whereas the country-level average covers a variety of forests, also including land at the very initial stages of regeneration as well as over-mature stands with low stocking. Our comparison suggested that young dense forest stands composed of European hornbeam may make a distinguished contribution to the total biomass (or carbon) stock of the country.

Our findings further indicate that alongside increasing stem contribution with the mean diameter $D_0$, the same trend was recorded for branches. On the other hand, the opposite trend was recorded for roots. Therefore, decreasing ratios between belowground and aboveground biomass with increasing stand dimensions can be concluded for our plots. Many authors (e.g., [34–37]) used a root to shoot ratio, i.e., the ratio between the belowground and aboveground biomass for a variety of purposes, for instance to link the biomass allocation response to specific growth conditions or environmental stress. Using worldwide data, Ledo et al. [37] showed that if considering the tree level, the ratio decreased in small trees (approximately up to 8 cm in diameter at breast height; DBH), then it stabilized and slightly increased in large trees. Hence, the decreasing tendency in the root to shoot ratio from worldwide meta-analyses considering small tree diameters conform with our findings from the young growth stages of European hornbeam presented here.

Rather surprisingly, the contribution of leaves to tree biomass along the examined mean diameter range remained nearly constant. As for the proportion of foliage to total biomass, its relevance is alongside other aspects linked to the carbon lifespan in forests. Whereas woody parts usually accumulate carbon during the whole tree life, leaves in deciduous species follow one-year turnover cycle. Therefore, leaf carbon is annually transported from tree crowns to the ground where it is continuously decomposed (e.g., [38,39]). Our estimates from young hornbeam stands showed that nearly 10% of the actual tree carbon stock is annually transported from tree crowns to the ground. This represented a quantity of about 6.0 tons of carbon per hectare at our plot with the thickest trees.

At the end of this section, we would like to explain the main reason for implementing diameter $D_0$ as a predictor in all biomass models. Our previous works [16,25,26,40], as well results from other authors (e.g., [41,42]), proved that a stem diameter (either $D_0$ or DBH) is nearly always the best independent variable for modelling tree biomass. The results also showed that tree height is a worse predictor than stem diameter; at the same time, adding tree height to stem diameter as a second predictor improves models only negligibly. Nevertheless, our current models expressing the dependence of mean height on mean diameter provides potential users with a chance to calculate stand biomass stock also using tree height as an alternative approach.

### 4.2. Importance of Leaf Traits

Leaf quantity and leaf traits are extremely important since foliage is an executor of photosynthesis, which is a principal process necessary for tree existence, growth and development. The SLA is considered an important indicator of physiological processes [43] since it links plant carbon and water cycles and can provide information on the spatial variation of photosynthetic capacity and leaf nitrogen content [44]. In addition, the SLA gives ideas about growth strategies at the ontogenetic as well as the intra- and inter-species levels [21]. Our results showed that the site-specific SLA of European hornbeam varied between 46.4 and 80.4 cm$^2$ per g of foliage dry mass. At the same time, the plot-level LAR was between 4.0 and 10.6 cm$^2$ per g of total plant dry mass. Our previous study [24], focusing on SLA at the tree level, showed that European hornbeam had lower values for SLA and LAR than three other broadleaved species (*Acer pseudoplatanus* L., *Betula pendula* Roth and *Populus tremula* L.).

The results indicate the high photosynthetic efficiency of European hornbeam leaves. At the same time, we also found high growth efficiency (GE) for hornbeam that is a physiological characteristic expressing a woody biomass increment per leaf area unit [45]. If we consider our model between mean plot diameter and LAR, increasing GE with increasing mean stem base diameter can be concluded. This is probably because of the limited growth potential of the woody parts in small trees due to the low number of cells in vascular cambium, which are the only producers of new woody cells and hence determine the radial increment [17]. In general, leaves are key determinants of the tree growth and development, and at the same time they are controlled by a variety of external factors [46].

Furthermore, we modelled the LAI using the mean plot diameter at the stem base. Eventually, a potential user could combine this dependence with the relationship between mean plot diameter and mean height if tree height was used as a predictor. The maximum LAI value (slightly over 6.0 m$^2$ per m$^2$) was estimated for a plot with a mean diameter of 35 mm. Here, we must point out that the observed values for the LAI come from dense stands with a full canopy. The information about the LAI in young forest stands is generally very sparse. Previously, we modelled the LAI in young common aspen stands [47]. The aspen model showed that a stand with a mean diameter of 35 mm had an LAI of around 6.0 m$^2$ per m$^2$, which coincides with the value from our model for European hornbeam. The aspen model showed that the value of the LAI further increased with the mean stand diameter; thus, the LAI approached a value of 12.0 m$^2$ per m$^2$ in a stand with a mean diameter of 100 mm. Therefore, we assume that the LAI of European hornbeam would also increase with a higher mean stand diameter outside the range covered in our current work.

Here, we would like to refer to the importance of the LAI in forest ecosystem research. The LAI quantifies the leaf area in an ecosystem and is a crucial variable for understanding physiological processes such as photosynthesis, respiration and precipitation interception [48–50]. Moreover, the global climate change research community has recognized the LAI as a fundamental attribute of global vegetation together with some other especially climate-related variables [51]. Therefore, developing precise and relatively cheap methods for LAI estimation is a serious task for scientists. For instance, the review by Fang et al. [52] provided a comprehensive analysis of LAI estimation from field measurements as well as using remote sensing estimation methods. They explained that, besides the implementation of models based on allometric relations, especially those using stem diameter as a predictor, remote sensing tools could also bring satisfying estimates of the LAI. Similarly, our works showed that the LAI could be precisely predicted using the mean stand diameter at stem base. On the other hand, our final model further illustrated that the LAI can be a good predictor for estimating total tree biomass. Thus, we can expect that remote sensing methods including the newest means, such as UAV and LiDAR, will provide reliable values for the LAI in the near future. In such cases, total tree biomass (sequestered carbon) may be estimated using models expressing biomass quantity based on the LAI.

Finally, we would like to point out that European hornbeam is affected, as other forest tree species are, by climate change. It seems that climate change would influence the species mostly in the positive way—in the form of the expansion of its range towards the north and to higher altitudes [53]. Similarly, in Slovakia, a continuous expansion of European hornbeam at the expense of oaks but also of European beech and other broadleaved hardwood and softwood species has already been observed [4]. The forest monitoring in Slovakia showed that the share of European hornbeam in tree species composition (based on the number of trees) increased from 5.2 to 8.9% between the years 2006 and 2016 [4]. This kind of phenomenon might endanger tree species diversity and consequently the sustainability of forest ecosystems [54]. Therefore, European hornbeam should be subjected to more intensive research than in the past.

## 5. Conclusions

Our work, focusing on young European hornbeam stands, brings new mathematical models for estimating the biomass of tree components at the stand level and LAR and LAI values by using mean diameter at stem base (or possibly tree height) as a predictor. We showed that the mean plot diameter is a very useful independent variable even in dense (number of trees can reach a hundred thousand individuals per ha) young hornbeam stands originating from natural seed regeneration. Moreover, we found that the LAI can be a relevant predictor of the total tree biomass. Since determining the LAI in forest stands has recently been a very profuse object of ongoing remote sensing research, we assume that in the future it can be implemented as a main or supplementary predictor of biomass using the data from remote sensing sources. We filled a scientific gap with our current models, since allometric relationships for young European hornbeam stands were completely missing and those from Western Europe were constructed only at the tree level and omitted the belowground parts. Moreover, the models could be useful for estimating biomass and foliage characteristics in young hornbeam stands in the countries of Central Europe, especially in the regions of the Carpathian arch.

An important output of our work focusing on young hornbeam stands, besides others, is the knowledge regarding the proportion of leaves to total tree biomass. Its relevance is linked, for instance, to carbon sequestration in forests. Moreover, our results showed that young dense forest stands composed of European hornbeam make a distinguished contribution to the total biomass (or carbon) stock in the Slovak forests when considering the national level. Our results based on the LAR brought the knowledge that European hornbeam leaves must be very efficient in photosynthesis per surface area. Our model expressing the relationship between mean plot diameter and LAR indicated that the GE

increased with increasing mean diameter at stem base. This is probably because of the limited growth potential of the woody parts in small trees due to the low number of cells in the vascular cambium of woody parts (i.e., branches, stem and roots).

Finally, we would like to point out that climate change would most probably affect European hornbeam in a positive way, resulting in the expansion of its range towards the north and to higher altitudes. Due to this, more scientific attention should be paid to this species than in the past. The research should especially focus on production and physiological and ecological aspects of European hornbeam under the ongoing climate change but also on the competitive relationship of European hornbeam with other tree species in mixed forest stands. Research activities should lead to propositions of forest management approaches with a priority to support close-to-nature species composition and high biodiversity.

**Author Contributions:** Conceptualization, B.K.; Data Curation, V.M., V.Š. and J.P.; Funding Acquisition, B.K., J.P. and V.Š. Investigation, B.K., V.Š. and J.P.; Methodology, V.M. and B.K.; Visualization, V.Š.; Supervision, B.K.; Writing—Original Draft Preparation, B.K., and K.M.; Writing—Review and Editing, all authors. All authors have read and agreed to the published version of the manuscript.

**Funding:** This research was funded by grant "EVA4.0", No. CZ.02.1.01/0.0/0.0/16_019/0000803 financed by OP RDE as well as within the projects APVV-18-0086, APVV-19-0387 and APVV-20-0168 from the Slovak Research and Development Agency.

**Institutional Review Board Statement:** Not applicable.

**Informed Consent Statement:** Not applicable.

**Conflicts of Interest:** The authors declare no conflict of interest.

## Appendix A

**Table A1.** Main characteristics of 200 sampled European hornbeam trees in Slovakia (adopted from Pajtík et al. [16]).

| Tree Characteristic | Mean | SD | Min. | Max. | 25% | 75% |
|---|---|---|---|---|---|---|
| Diameter $D_0$ (mm) | 17.84 | 13.94 | 0.90 | 81.20 | 7.90 | 23.3 |
| Height (m) | 2.68 | 1.83 | 0.07 | 7.56 | 1.10 | 3.88 |
| Leaf biomass (g) | 31.18 | 59.22 | 0.03 | 346.34 | 1.78 | 35.67 |
| Branch biomass (g) | 56.10 | 138.84 | 0.01 | 935.15 | 1.65 | 43.15 |
| Bark biomass (g) | 34.82 | 59.61 | 0.01 | 391.55 | 2.67 | 35.15 |
| Stem under bark biomass (g) | 263.27 | 596.14 | 0.03 | 4038.19 | 9.60 | 191.30 |
| Root biomass (g) | 78.40 | 185.46 | 0.04 | 1473.50 | 5.83 | 60.50 |
| Aboveground biomass (g) | 346.45 | 713.01 | 0.06 | 4329.52 | 12.70 | 284.95 |
| Tree biomass (g) | 414.91 | 858.13 | 0.10 | 5399.22 | 19.44 | 341.31 |

**Table A2.** Allometric models for European hornbeam tree components based on diameter at stem base $D_0$ (mm) and tree height (m), adopted from Pajtík et al. [16]. The formula is as follows: $W_i = e^{(b_0 + b_1 \ln D_0 + b_2 \ln H)} \lambda$, where: $W_i$ is biomass (weight in g) of $i$th tree component, $D_0$ is stem base diameter (mm), $H$ is tree height (m), $b_0$, $b_1$ and $b_2$ are regression coefficients and $\lambda$ is correction factor. Other abbreviations in the table are: S.E.—standard error, $p$—$p$ value, $R^2$—coefficient of determination, MSE—mean square error, S.D.—standard deviation.

| Tree Component | $b_0$ | S.E. | $p$ | $b_1$ | S.E. | $p$ | $b_2$ | S.E. | $p$ | $R^2$ | MSE | $\lambda$ | S.D. |
|---|---|---|---|---|---|---|---|---|---|---|---|---|---|
| Leaves | −5.423 | 0.366 | <0.001 | 3.008 | 0.177 | <0.001 | −0.599 | 0.152 | <0.001 | 0.893 | 0.463 | 1.211 | 0.696 |
| Branches | −5.514 | 0.291 | <0.001 | 2.893 | 0.140 | <0.001 | 0.029 | 0.123 | 0.816 | 0.946 | 0.296 | 1.143 | 0.594 |
| Bark | −2.026 | 0.106 | <0.001 | 1.396 | 0.051 | <0.001 | 0.965 | 0.045 | <0.001 | 0.971 | 0.109 | 1.055 | 0.353 |
| Stem under bark | −1.486 | 0.106 | <0.001 | 1.841 | 0.051 | <0.001 | 0.896 | 0.044 | <0.001 | 0.993 | 0.041 | 1.021 | 0.220 |
| Roots | −3.348 | 0.174 | <0.001 | 2.424 | 0.084 | <0.001 | −0.114 | 0.073 | 0.121 | 0.991 | 0.041 | 1.018 | 0.183 |
| Whole tree | −1.445 | 0.121 | <0.001 | 2.165 | 0.058 | <0.001 | 0.433 | 0.050 | <0.001 | 0.990 | 0.049 | 1.024 | 0.226 |

**Table A3.** Basic characteristics of European hornbeam trees measured at 30 plots.

| Site Code | Site Name | Plot (A–C) | Diameter $D_0$ (mm) | | Tree Height (m) | | Basal Area * ($cm^2$ per $m^2$) | Tree Density (pcs $m^{-2}$) |
|---|---|---|---|---|---|---|---|---|
| | | | Mean | SD | Mean | SD | | |
| | Píla | A | 13.85 | 5.43 | 2.06 | 0.78 | 18.73 | 11 |
| 1 | Píla | B | 33.44 | 17.29 | 4.76 | 1.57 | 101.98 | 9 |
| | Píla | C | 29.05 | 18.18 | 3.38 | 1.84 | 32.82 | 7 |
| | Rudica | A | 13.45 | 15.35 | 1.84 | 1.14 | 49.40 | 15 |
| 2 | Rudica | B | 10.01 | 5.68 | 1.42 | 0.93 | 14.49 | 14 |
| | Rudica | C | 11.21 | 5.70 | 2.01 | 0.94 | 20.47 | 17 |
| | Antol | A | 26.91 | 14.75 | 4.61 | 1.36 | 93.63 | 13 |
| 3 | Antol | B | 30.70 | 13.34 | 4.59 | 1.37 | 94.80 | 11 |
| | Antol | C | 27.23 | 15.33 | 4.08 | 1.46 | 70.20 | 9 |
| | Dol. Breziny | A | 25.91 | 11.96 | 4.58 | 1.41 | 52.32 | 8 |
| 4 | Dol. Breziny | B | 26.82 | 13.99 | 4.73 | 1.69 | 47.56 | 7 |
| | Dol. Breziny | C | 25.01 | 12.45 | 4.71 | 1.51 | 81.58 | 13 |
| | Šariny | A | 9.94 | 6.96 | 1.44 | 1.05 | 17.78 | 15 |
| 5 | Šariny | B | 13.93 | 9.91 | 1.99 | 1.30 | 19.31 | 8 |
| | Šariny | C | 12.49 | 6.20 | 2.15 | 1.07 | 26.62 | 18 |
| | Hor. Breziny | A | 3.99 | 1.66 | 0.45 | 0.18 | 34.68 | 237 |
| 6 | Hor. Breziny | B | 3.88 | 1.81 | 0.44 | 0.18 | 26.12 | 183 |
| | Hor. Breziny | C | 4.36 | 1.95 | 0.42 | 0.21 | 24.00 | 134 |
| | Cerovo | A | 13.64 | 7.13 | 1.89 | 0.99 | 40.14 | 22 |
| 7 | Cerovo | B | 17.73 | 9.52 | 2.85 | 0.98 | 61.53 | 19 |
| | Cerovo | C | 12.88 | 5.24 | 2.22 | 0.75 | 56.17 | 37 |
| | Soroška | A | 14.88 | 8.50 | 1.64 | 0.70 | 23.32 | 10 |
| 8 | Soroška | B | 16.71 | 7.14 | 2.26 | 0.58 | 32.88 | 13 |
| | Soroška | C | 18.18 | 8.57 | 2.26 | 0.77 | 29.11 | 9 |
| | Budimír | A | 30.55 | 16.26 | 4.45 | 1.23 | 86.20 | 9 |
| 9 | Budimír | B | 24.79 | 11.12 | 3.95 | 1.06 | 67.98 | 12 |
| | Budimír | C | 27.78 | 16.07 | 4.12 | 1.43 | 99.81 | 12 |
| | Zubné | A | 22.05 | 10.83 | 3.57 | 1.30 | 63.08 | 13 |
| 10 | Zubné | B | 21.52 | 11.21 | 3.41 | 1.23 | 54.13 | 12 |
| | Zubné | C | 18.86 | 8.45 | 3.06 | 0.89 | 61.46 | 18 |

Explanatory note: * Basal area at stem base, i.e., the sum of tree sections per $m^2$ of stand area calculated from diameters measured on stem bases (i.e., at the ground level).

**Table A4.** Performance measures of all regression models: NSE (Nash–Sutcliffe efficiency), RMSE (root mean squared error), PBIAS (percent bias), IOA (index of agreement), AIC (Akaike information criterion) and Bayesian information criterion (BIC).

| Related Variables | (Equation) | NSE | RMSE | PBIAS | IOA | AIC | BIC |
|---|---|---|---|---|---|---|---|
| H vs. $D_0$—tree level | (1) | 0.88 | 0.63 | 0.03 | 0.97 | 2680 | 2701 |
| mean H vs. mean $D_0$—plot level | (1) | 0.95 | 0.32 | 0.38 | 0.99 | 23 | 29 |
| leaf biomass vs. mean $D_0$ | (2) | 0.86 | 123.47 | 2.54 | 0.96 | 368 | 372 |
| branch biomass vs. mean $D_0$ | (2) | 0.91 | 181.88 | 0.57 | 0.98 | 390 | 394 |
| bark biomass vs. mean $D_0$ | (2) | 0.83 | 126.52 | 0.62 | 0.95 | 369 | 373 |
| root biomass vs. mean $D_0$ | (2) | 0.85 | 214.82 | 2.03 | 0.96 | 400 | 404 |
| stem biomass vs. mean $D_0$ | (2) | 0.89 | 744.81 | −0.20 | 0.97 | 472 | 476 |
| total biomass vs. mean $D_0$ | (2) | 0.89 | 1342.86 | 0.69 | 0.97 | 506 | 510 |
| LA vs. LW—leaf level | (3) | 0.78 | 97.57 | −0.48 | 0.93 | 13,181 | 13,196 |
| LA vs. LW—stand level | (4) | 0.79 | 26.62 | $-1.97 \times 10^{-16}$ | 0.94 | 100 | 101 |
| LAR vs. $D_0$—tree level | (5) | 0.39 | 2.67 | −0.35 | 0.73 | 751 | 760 |
| LAR vs. $D_0$—plot level | (5) | 0.70 | 0.94 | 0.10 | 0.91 | 85 | 89 |
| LAI vs. $D_0$—plot level | (6) | 0.77 | 0.93 | 2.84 | 0.94 | 87 | 91 |
| total biomass vs. LAI—plot level | (7) | 0.94 | 921.62 | $-6.03 \times 10^{-16}$ | 0.99 | 501 | 505 |

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
