# Peer review of "Stand-Level Biomass and Leaf Trait Models for Young Naturally Regenerated Forests of European Hornbeam"

_forests, doi:10.3390/f14061084_

Round 1
Reviewer 1 Report
The manuscript "Biomass and LAI in Dense Young Stands of European Hornbeam Originating from Natural Generative Regeneration" investigated stand-level allometric relations of biomass stock in tree components. The results provided a good reference for the phenology of European hornbeam (Carpinus betulus L.). It is worthy of publication after a Minor Revision.
(1) The title is superficial and the core innovation is hard to embody.
(2) The Abstract section should be refined, instead of only listing the correlation results. The mechanism behind these phenomena should also be discussed.
(3) Some information on “stand selection and tree sampling” was lacking, e.g., sampling time, parallel samples, and selection of sampling trees.
(4) The statistical criteria in this study were not enough to evaluate the model performance. Other statistical criteria should be used, e.g., NSE, PBIAS, IoA. The NSE was applied to normalize the residual variance between the observed and simulated data. The PBIAS was used to evaluate the difference between the mean observed and simulated data.
(5) The conclusion should be further refined. It should directly underscore the major and novelty of the findings, and some key data can be presented.
Author Response
We would like to express thanks to all reviewers for the comments and suggestions. We considered them and reviewed the text based on them.
Reviewer 2
The manuscript "Biomass and LAI in Dense Young Stands of European Hornbeam Originating from Natural Generative Regeneration" investigated stand-level allometric relations of biomass stock in tree components. The results provided a good reference for the phenology of European hornbeam (Carpinus betulus L.). It is worthy of publication after a Minor Revision.
(1) The title is superficial and the core innovation is hard to embody.)
Authors: The title was modified.
(2) The Abstract section should be refined, instead of only listing the correlation results. The mechanism behind these phenomena should also be discussed.
Authors: The text of the Abstract was changed accordingly.
(3) Some information on stand selection and tree sampling” was lacking, e.g., sampling time, parallel samples, and selection of sampling trees.
Authors: We added some more information about procedures of tree selection and sampling time. Moreover, we referred to the monograph by Pajtík et al.(2018), which shows very detailed procedures of stand and tree selection as well as the field and laboratory work.
(4) The statistical criteria in this study were not enough to evaluate the model performance. Other statistical criteria should be used, e.g., NSE, PBIAS, IoA. The NSE was applied to normalize the residual variance between the observed and simulated data. The PBIAS was used to evaluate the difference between the mean observed and simulated data.
Authors: We agree with the suggestion. Therefore, we added some more statistical criteria to evaluate the models (see separate Table A4 in Attachment).
(5) The conclusion should be further refined. It should directly underscore the major and novelty of the findings, and some key data can be presented.
Authors: We agree with the reviewer. The Conclusions section was enriched with some more findings of the work with adequate interpretation.
Reviewer 2 Report
The contribution is interesting because it shows some relationships between leaf indexes and average diameter at stump height in very young European hornbean stands. Besides that, it is interesting to show the big amount of biomass and CO2 that these stands - regenerated naturally from seeds - can grow in a short time.
But there are some confusing information which must be presented in a clearer way and there is also a lack of relevant data that has to be completed in order to gain significance:
- First of all, the summary must indicate that the analyzed stands are not only "young" but "younger than 10 years old". Also the authors must precise, including in the Abstract, that the "mean stand diameter" does not refer to standard DBH but to the "average stand diameter at the stump height".
- All the references to biomass weight must indicate the moisture content - on humid or dry basis - in order to permit comparisons with other references. That aspect must be corrected prior to publish the article.
- The terms "natural generative regeneration" is not clear to me, I think that it would be better using "natural regeneration from seeds" or simply "natural regeneration", as it is defined by UK's Forest Research as "the process by which woodlands are restocked by trees that develop from seeds that fall and germinate in situ."
- In page 4, the reference to "mean basal area"... calculated from the average diameter at stump heigth can be misunderstood, as basal area is defined as "the cross-sectional area of trees at breast height (1.3m or 4.5 ft above ground)", so the authors must refer to "the sum of tree sections per hectare at stump height" instead of "basal area".
- The Table A2 should express the density in trees per hectare, much better than pcs. per square meter, as it is a more standard unit to allow the comparison with another forest studies. The range of densities and its average value must be also indicated in the text, not only in the Table, even in the abstract, to ease the comprension if the study object to the readers.
When these lacking or confusing pieces of informatios were corrected, the manuscript will deserve its publication.
The English redaction should be revised by an English native speaker with knowledge about forest sciences
Author Response
We would like to express thanks to all reviewers for the comments and suggestions. We considered them and reviewed the text based on them.
Reviewer 3
- First of all, the summary must indicate that the analyzed stands are not only "young" but "younger than 10 years old". Also the authors must precise, including in the Abstract, that the "mean stand diameter" does not refer to standard DBH but to the "average stand diameter at the stump height".
Authors: OK, we added (modified) the information in the Abstract section according to the suggestions from the reviewer.
- All the references to biomass weight must indicate the moisture content - on humid or dry basis - in order to permit comparisons with other references. That aspect must be corrected prior to publish the article.
Authors: The weight of the biomass is always under the conditions of zero moisture content in tree material. We added this information in the Material and methods section.
- The terms "natural generative regeneration" is not clear to me, I think that it would be better using "natural regeneration from seeds" or simply "natural regeneration", as it is defined by UK's Forest
Research as "the process by which woodlands are restocked by trees that develop from seeds that fall and germinate in situ."
Authors: We agree, this was modified in the title and also in the entire text of the manuscript.
- In page 4, the reference to "mean basal area"... calculated from the average diameter at stump heigth can be misunderstood, as basal area is defined as "the cross-sectional area of trees at breast height (1.3m or 4.5 ft above ground)", so the authors must refer to "the sum of tree sections per hectare at stump height" instead of "basal area".
Authors: OK, this fact was modified in the entire text of the manuscript.
- The Table A2 should express the density in trees per hectare, much better than pcs. per square meter, as it is a more standard unit to allow the comparison with another forest studies. The range of densities and its average value must be also indicated in the text, not only in the Table, even in the abstract, to ease the comprension if the study object to the readers.
Authors: Thanks for the suggestion. Although we agree that the standard expression of stand density is in trees per hectare, upscaling to per hectare basis in young stands is rather questionable due to the great variability of density at a microsite level. The data were collected at very small plots (much smaller than usually established in mature stands), which do not necessarily represent the situation at a larger scale, where the actual number per hectare may be different from the calculated figures. In our case, upscaling may bring systematically skewed and higher values than in reality, because we deliberately focused on fully stocked parts of young stands, while in reality some stand parts could have low tree density due to the unsuccessful regeneration. Therefore, we consider the conversion to m2 better than to hectare, since the area of 1m2 is very similar to the real size of surveyed area. Nevertheless, we compared our results converted to a per hectare basis with other works in the Discussion section.
When these lacking or confusing pieces of information were corrected, the manuscript will deserve its publication.
Authors: Thank you for the positive evaluation!
Round 2
Reviewer 2 Report
With the required changes and/or clarifications, the article does deserve its publication in Forests